# Salivary Endocannabinoid Profiles in Chronic Orofacial Pain and Headache Disorders: An Observational Study Using a Novel Tool for Diagnosis and Management

**DOI:** 10.3390/ijms232113017

**Published:** 2022-10-27

**Authors:** Shimrit Heiliczer, Asaf Wilensky, Tal Gaver, Olga Georgiev, Sharleen Hamad, Alina Nemirovski, Rivka Hadar, Yair Sharav, Doron J. Aframian, Joseph Tam, Yaron Haviv

**Affiliations:** 1Department of Oral Medicine, Sedation and Maxillofacial Imaging, Hadassah Medical Center, Faculty of Dental Medicine, The Hebrew University of Jerusalem, Jerusalem 91120, Israel; 2Department of Periodontology, Hadassah Medical Center, Faculty of Dental Medicine, The Hebrew University of Jerusalem, Jerusalem 9112001, Israel; 3In Partial Fulfillment of DMD Requirements, Hadassah Medical Center, Faculty of Dental Medicine, The Hebrew University of Jerusalem, Jerusalem 91120, Israel; 4Obesity and Metabolism Laboratory, The Institute for Drug Research, School of Pharmacy, Faculty of Medicine, The Hebrew University of Jerusalem, Jerusalem 91120, Israel

**Keywords:** endocannabinoids, anandamide, 2-AG, chronic pain, orofacial pain, neuropathic pain, migraine, saliva

## Abstract

The endocannabinoid system is involved in physiological and pathological processes, including pain generation, modulation, and sensation. Its role in certain types of chronic orofacial pain (OFP) has not been thoroughly examined. By exploring the profiles of specific salivary endocannabinoids (eCBs) in individuals with different types of OFP, we evaluated their use as biomarkers and the influence of clinical parameters and pain characteristics on eCB levels. The salivary levels of anandamide (AEA), 2-arachidonoyl glycerol (2-AG), and their endogenous breakdown product arachidonic acid (AA), as well as the eCB-like molecules N-palmitoylethanolamide (PEA) and N-oleoylethanolamide (OEA), were assessed in 83 OFP patients and 43 pain-free controls using liquid chromatography/tandem mass spectrometry. Patients were grouped by diagnosis: post-traumatic neuropathy (PTN), trigeminal neuralgia (TN), temporomandibular disorder (TMD), migraine, tension-type headache (TTH), and burning mouth syndrome (BMS). Correlation analyses between a specific diagnosis, pain characteristics, and eCB levels were conducted. Significantly lower levels of 2-AG were found in the TN and TTH groups, while significantly lower PEA levels were found in the migraine group. BMS was the only group with elevated eCBs (AEA) versus the control. Significant correlations were found between levels of specific eCBs and gender, health-related quality of life (HRQoL), BMI, pain duration, and sleep awakenings. In conclusion, salivary samples exhibited signature eCBs profiles for major OFP disorders, especially migraine, TTH, TN, and BMS. This finding may pave the way for using salivary eCBs biomarkers for more accurate diagnoses and management of chronic OFP patients.

## 1. Introduction

Chronic orofacial pain (OFP) is a debilitating condition that is associated with the structures innervated by the trigeminal nerve (head, face, and intraoral structures). It is one of the most common pain conditions, with a reported prevalence of 7–11% in the general population [1]. Currently, the presumed underlying causes of chronic OFP are classified as musculoskeletal, neuropathic, or neurovascular. The complex histories, pathophysiology, and associated psychosocial co-morbidities (e.g., depression and anxiety) make the diagnosis and management of OFP disorders challenging [2]. The difficulty is compounded by the lack of clarity regarding the pathophysiological mechanisms and etiology of these pain disorders [3].

Due to the nature of pain as a sensory experience that cannot be directly quantified or measured [4], subjective ratings play a vital role in pain diagnosis and treatment. These ratings are complicated by profound individual differences in sensitivity [5]. Therefore, pain assessment using biomarkers related to pain mechanisms may provide the objective data we are currently lacking [4]. Indeed, the search for pain biomarkers focuses on identifying objective, measurable correlates to the neurobiological processes that cause these conditions in order to base diagnosis and treatment on the underlying pathophysiological mechanisms rather than symptomatology [5].

Sensation to the orofacial tissues, such as teeth, facial skin, TMJ, and adjacent musculature, is mainly supplied by branches of the trigeminal (V) nerve [6]. Pain pathways are part of this complex sensory system. Input regarding noxious stimuli is transmitted from nociceptors by primary afferent Aδ and C-fibers. These fibers have cell bodies in the trigeminal ganglion and synapse with neurons in the V brainstem complex, mainly in the subnucleus caudalis. Various neurotransmitters, such as glutamate and calcitonin gene-related peptide (CGRP), are released via signal transduction. Projection neurons from the subnucleus caudalis ascend in the ventral trigeminothalamic tract to the ventral posterolateral nuclei of the thalamus. Finally, the information is transmitted to the somatosensory cortex and brain areas involved in memory and affective aspects of pain (amygdala, hypothalamus, etc.) [7,8,9]. The development of chronic pain is associated with synaptic plasticity, which causes changes in various areas of the CNS that modulate pain [7].

In this context, recent research found that the endogenous cannabinoid (endocannabinoid (eCB)) system is essential in pain pathophysiology [10], and there is evidence that it has a critical modulatory role in nociception [11]. The endocannabinoid system (ECS) operates in the CNS and its periphery and has three principal components: (1) “classical” (CB1, CB2) and “non-classical” (transient receptor potential vanilloid 1 (TRPV1), G-protein-coupled receptor 55 (GPR55), and peroxisome proliferator-activated receptors (PPARs)) cannabinoid receptors; (2) endogenous ligands (*N*-arachidonoylethanolamide (anandamide) (AEA) and 2-arachidonoyl glycerol (2-AG)); and (3) enzymes, which are responsible for the biosynthesis/inactivation of the ligands. Furthermore, AEA is a member of the *N*-acylethanolamines (NAEs) family, which includes the eCB-like compounds *N*-palmitoylethanolamide (PEA) and *N*-oleoylethanolamide (OEA) [12]. 

eCBs are synthesized and released locally by enzymatic cleavage of membrane phospholipids in response to physiological and pathological stimuli [13]. *N*-acylphosphatidylethanolamine-phospholipase D (NAPE-PLD) catalyzes AEA biosynthesis, whereas 2-AG is catalyzed by sn-1-specific diacyl-glycerol lipase (DAGL). The released eCBs are retrieved by a membrane transporter. AEA is degraded by fatty acid amide hydrolase (FAAH), whereas monoglyceride lipase (MAGL) is the main 2-AG hydrolase [14]. 2-AG is a full agonist of CB1 and CB2 receptors, whereas AEA shows slight selectivity for CB1 over CB2 [11]. AEA also acts on other receptors implicated in pain processing, such as TRPV1 and PPARs [13]. ECS components are expressed ubiquitously throughout pain-processing pathways [11], which balance GABAergic inhibitory activity and glutamatergic excitatory activity by eCBs released from postsynaptic neurons that stimulate presynaptic CB1 [14]. Based on their neuromodulatory role [12], it is reasonable to consider eCBs as reliable biomarkers for chronic pain conditions. 

Whereas blood serum/plasma is the most frequent source of measurable biomarkers, saliva has many advantages over blood as a collectible bio-fluid, such as being easy to collect in a non-invasive manner and safer to handle. Many substances enter saliva from the blood via intercellular spaces due to transcellular or paracellular diffusion, and most substances found in the blood are also present in saliva, reflecting physiological and pathological states, making it a useful source of biomarkers for pain research [15].

A recent study conducted by our group demonstrated significantly reduced levels of salivary eCB in chronic OFP disorders, which were categorized by etiology (neuropathic, neurovascular, and musculoskeletal) [16]. This avenue of research was broadened in the current study with the following objectives: (I) to investigate the quality and quantity of specific eCB compounds in distinct chronic OFP disorders and to evaluate their efficacy as measurable biomarkers of these pain disorders, and (II) to evaluate the influence of different clinical parameters and pain characteristics on salivary eCB levels in specific chronic OFP disorders.

## 2. Results

A total of 126 individuals participated in the current study, 83 of which had OFP, including TMD (25.3%), migraine (28.9%), TTH (6%), PTN (13.4%), TN (13.4%), and BMS (8.5%). A total of 4.8% were defined as “others”. Figure 1 summarizes the cohort distribution. 

Of the 126 participants forming the cohort, 83 (66%) had OFP, while 43 (34%) were pain-free and formed the control group. Of the OFP group 21 (25.3%) had TMD, 24 (28.9%) had migraine, 5 (6%) had TTH, 11 (13.4%) had PTN, 11 (13.4%) had TN, 7 (8.5%) had BMS, and the remaining 4 (4.8%) were defined as “others”.

The primary headaches group represents the sum of migraine (including orofacial migraine) and TTH, while the neuropathic group included PTN, TN, and BMS. 

Table 1 summarizes the significant differences between TMD, primary headaches (migraine and TTH), the neuropathic pain group (including PTN, TN, and BMS), and the control group. Due to the non-significant differences in salivary eCB levels in the TMD group compared with the controls, we focused on the migraine and neuropathic pain groups.

Table 2 summarizes the salivary eCB levels in the pain groups that were significantly different from the controls. 

Table 3 summarizes the patient and pain characteristics with significant correlations to salivary eCBs in PTN, TN, and migraine. Gender, BMI, VPS, HRQoL, sleep awakening, burning and pressing pain quality, and pain onset were found to be significantly related to certain eCB levels. 

Considering the vast amount of data collected and the many comparisons performed between the clinical factors and patient pain characteristics for each of the five measured eCBs, we decided to only present the statistically significant results. It is important to note that for some of the diagnoses, e.g., BMS and TTH, there were not enough patients for within-group comparisons.

## 3. Discussion

The limited understanding of the etiology and pathogenesis of OFP disorders, the subjective nature of current pain assessments, and the limited efficacy of existing treatment options highlight the urgent need for objective data to assist with chronic pain evaluation and management. The identification of mechanistic biomarkers is crucial since it would not only improve our understanding and ability to diagnose pain disorders accurately but also facilitate the development of disease-modifying drugs [3]. Over the past few decades, an endogenous system with previously unknown anti-nociceptive properties was revealed, namely, the ECS [11]. Studies found that eCBs produced in response to high levels of stimulation cause an analgesic effect via negative feedback and inhibition of the transmission of pain signals [11]. Accordingly, the current study aimed to determine whether different types of chronic OFP disorders alter salivary levels of eCBs (2-AG, AEA), their endogenous breakdown product (AA) levels, and eCB-like compounds (PEA, OEA).

Our data demonstrated significantly lower salivary PEA levels in the migraine group compared with the controls. No other significant differences between these groups were found. PEA, an endogenous fatty acid amide signaling molecule, is synthesized on demand as a protective response to tissue injury or stress as part of homeostatic mechanisms. It has anti-inflammatory, pain relieving, and neuroprotective actions [19]. PEA’s direct targets are the PPARα and GPR55 receptors. PPARα is a transcriptional factor that promotes the expression of genes with anti-inflammatory activity and GPR55 regulates neuroinflammation [19].

One of the most important anti-inflammatory effects of PEA is the inhibition of mast cell activation [20], which is critical in the development of inflammation in migraine. Furthermore, the administration of ultra-micronized PEA caused significant pain relief and a reduction in the number of migraine attacks in pediatric migraineurs [19]. 

Interestingly, in contrast to our findings, Sarchielli et al. [21] found that PEA was significantly higher, whereas AEA levels were lower in the cerebrospinal fluid (CSF) of chronic migraineurs compared with the controls. Since these dissimilarities were found in different biofluids (CSF versus saliva), and since PEA is hydrolyzed by various enzyme classes, namely, FAAH and *N*-acylethanolamine acid amidase (NAAA), which catabolize PEA and AEA at significantly different rates [22], it is possible that the differences in the quality and quantity of these enzymes in the examined fluids caused the disparity. Nonetheless, our study strengthened the evidence of a dysregulated ECS ‘tone’ with reduced eCB activity, which may play a role in migraines [14].

Our data also demonstrated low salivary 2-AG levels in the TTH group compared with the controls. Although TTH is the most prevalent type of headache, its pathophysiology remains unclear [23]. Interrelationships between peripheral and central mechanisms seemingly underlie TTH initiation [18]. Olesen [24] suggested that TTH occurs due to an interaction between the descending inhibitory system, which controls nociceptive brainstem neurons, and peripheral input from the vascular system, with mainly myofascial sources. In addition, input from the limbic structures (e.g., anxiety, depression) may reduce descending inhibitory function and produce more chronic TTH [23]. 2-AG has high affinity and potency regarding both CB1 and CB2 receptors. It is anti-nociceptive and is thought to participate in pain initiation. Hence, its reduced levels in TTH may contribute to the development of chronic pain disorders. Considering the wide distribution of cannabinoid receptors along central and peripheral pain pathways [13], it is reasonable to assume that 2-AG deficiency will affect both the peripheral and central mechanisms underlying TTH. Furthermore, women with depression had significantly lower circulating 2-AG levels [25]. Therefore reduced 2-AG may have an impact on limbic components with a role in TTH pathophysiology. 

We also found significantly lower levels of salivary 2-AG in the TN group. TN is a chronic neuropathic condition that affects one or more divisions of the trigeminal nerve [2]. Demyelination is the predominant theory of the cause of neuralgic pain [23]. Devor et al. [26] described the ignition hypothesis, whereby TN starts following damage to trigeminal axons in the nerve root or ganglion, often due to vascular compression of the nerve [2]. 2-AG stimulation of cannabinoid receptors may reduce excitatory neurotransmitter release and attenuate neuropathic pain symptoms [13]. Furthermore, 2-AG may have neuroprotective effects in response to harmful stimuli [27]. The abovementioned studies support 2-AG’s significant role in nerve damage and neuropathic pain prevention. Indeed, various studies reported increased eCB levels at central and peripheral nervous system sites in neuropathic pain conditions. This elevation may be caused by inbuilt protective mechanisms responding to a pathological condition [13]. We suggest that the lower salivary 2-AG levels in TN were due to a defect in this compensatory eCB mechanism. 

Surprisingly, we found an elevation in salivary AEA levels for BMS compared with the controls. No other significant differences between these groups were found. BMS is a chronic neuropathic pain disorder characterized by an oral mucosal burning sensation and is frequently associated with xerostomia and dysgeusia [28]. The etiology of BMS is unknown, and studies have associated it with inflammation and psychiatric disturbances [12]. Interestingly, immune-histochemical staining of tongue biopsies revealed significantly increased TRPV1, decreased CB1 receptor, and increased CB2 receptor expressions in the epithelium [29]. In addition, neurotrophic factors, which regulate TRPV1 expression, i.e., nerve growth factor and artemin, were also overexpressed in BMS [28]. The previously reported TRPV1 overexpression, together with our findings of increased AEA levels, may be related to BMS symptoms. This receptor, which is found peripherally in the nociceptive terminals of Aδ and C-fibers, conducts heat, spicy taste (capsaicin), and nociceptive signals, and its expression corresponds with hypersensitivity to noxious heat stimulation, such as to capsaicin experienced by those with BMS [29]. TRPV1 is an ionotropic cannabinoid receptor and is desensitized upon agonist exposure, where AEA is an agonist of this receptor [30]. However, AEA has low intrinsic efficacy at this receptor and is only a partial agonist in the trigeminal nerve [31]. The physiological implication of action as a partial agonist is attenuation of the effects of full agonists, whether exogeneous, e.g., capsaicin, or endogenous, e.g., *N*-arachidonoyl-dopamine. Additionally, the intrinsic efficacy of AEA on TRPV1 is lower than at the CB1 receptor, implying that in the presence of CB1, AEA action on TRPV1 may be attenuated [31]. AEA is also a partial agonist of CB1 and CB2, whereas 2-AG is a full agonist of these receptors [32]. Indeed, 2-AG is three times more potent than AEA, and effective attenuation of the functional activity of 2-AG was demonstrated in a pre-clinical trial when co-incubated with AEA [33]. 

Therefore, despite AEA being a known anti-nociceptive eCB and its stimulation of cannabinoid receptors producing analgesic effects, it may act as an antagonist in the presence of full cannabinoid agonists [34]. 

Taken together, we hypothesize that the significant elevation of salivary AEA in BMS with no concomitant elevations of full eCB agonists, such as 2-AG, reflects a faulty mechanism of the ECS in these patients. The elevated AEA accompanied by altered epithelial tongue cell eCB receptor expression [12] implies that altered eCB signaling is involved in BMS pathogenesis. Salivary AEA elevation may also play a role in the frequently reported BMS-related xerostomia [28]. This assumption is supported by in vivo and in vitro studies [35] showing that AEA decreased saliva secretion from the submandibular glands via CB1 and CB2 receptors. Kopach et al. [36] also elucidated CB receptor-mediated salivary regulation.

In addition to the correlations between specific pain disorder subgroups and salivary eCB levels, we also found a correlation between ECS activity and patient characteristics in three pain disorder groups: TN, PTN, and migraine. There were significantly lower salivary AA levels in women than men in the PTN group and this trend was also noted for OEA and PEA levels in the migraine group. These findings are consistent with Cupini et al. [37], who detected a higher activity of FAAH and the AEA transporter in the platelets of female migraine patients than in males. In rodents, the ECS within pain modulatory pathways demonstrated sexual dimorphism, with significant differences between males and females in 2-AG and AEA concentrations [38]. Furthermore, it was suggested that sex differences in eCB ‘tone’ develop early, with growing pre-clinical evidence that gonadal hormones influence the expression of cannabinoid receptors and ligands, as well as their affinity and efficacy at receptors in various brain regions [39]. These differences help to explain the well-documented higher prevalence of chronic pain disorders in females [38]. Nonetheless, further research is needed to investigate ECS gender dimorphism and to correlate this disparity with the manifestations of chronic pain.

Interestingly, our data indicated a significant negative correlation between the VPS and salivary OEA and PEA levels in the migraine group. This is congruent with previous studies, where chronic administration of PEA reduced pain behaviors and counteracted spinal neuronal hyper-excitability in murine models of persistent pain [40,41].

HRQoL reflects the perceptions and reactions of an individual to their health status and the non-medical aspects of their lives and assesses their psychological functioning and, to a smaller extent, physical functioning [42]. We found a significant positive correlation between the HRQoL score and salivary AA levels in the PTN group. Indeed, pre-clinical and human data demonstrate that eCB/CB1 receptor signaling regulates stress responses and moods. Thus, inhibition of this signaling can result in increased anxiety [38]. Lower circulating AEA levels are associated with higher anxiety measurements [43] and lower circulating 2-AG concentrations in women with depression. Taken together, we suggest that altered ECS activity in this chronic pain group may affect pain perception and psychological status.

We found that a BMI above 30 was significantly related to higher salivary 2-AG levels in the migraine group. This is consistent with previous studies, where levels of circulating 2-AG positively correlated with BMI, total body fat, and intra-abdominal adipose tissue [31,43]. This correlation was not found in the other pain subgroups and was possibly concealed by the dysregulated ECS activity in these patients that was discussed above.

In the TN group, a positive correlation was found between sleep awakenings and salivary AEA, OEA, and PEA levels. Significant OEA elevation was shown in human CSF after 24 h of sleep deprivation [44]. It is possible that OEA elevation has a neuroprotective role in sleep-deprived individuals via the activation of PPARα, yet the AEA levels were stable in this particular investigation [44]. Other studies demonstrated serum AEA elevations later in the day following a night of complete sleep deprivation [45]. Systemic administration of AEA significantly increased adenosine levels in the basal forebrain of rats and increased sleep [46]. These findings may imply an auto-regulatory role of AEA in sleep deprivation. PEA may be involved in sleep regulation since eight weeks of daily PEA supplementation reduced sleep onset time and improved cognition on waking [47]. Inconsistent with our findings are previous studies that identified an elevation in circulating 2-AG following restricted sleep [43]. This could be explained by the significantly lower salivary 2-AG levels found in the TN group compared to controls. 2-AG’s role in sleep deprivation was probably related to dysfunctional ECS activity in this chronic pain group.

Finally, our data demonstrated a significant positive correlation between pain duration and salivary AA levels in the TN group. This may have been due to pain-management-related symptom relief over time. This assumption is strengthened by the inverse correlation between pain intensity and salivary eCB levels [16].

There were some important limitations to this carefully designed study. First, we only measured salivary eCB levels from samples taken at a single point in time, gathering samples over more time points may reveal other important information, such as correlations between treatment outcomes and eCB levels or their correlation with long-term clinical symptoms. Second, there were several therapeutic treatments with different efficacies, which could have affected the salivary eCBs concentrations. Third, serum eCB levels were not measured in our study. Thus, a correlation between salivary and circulating eCBs could not be evaluated. Fourth, we had relatively small sample sizes for each subgroup, which limited the comparative statistical analysis.

## 4. Materials and Methods

### 4.1. Participants

The study was approved by the Ethical Committee of Hadassah Medical Center, request no. 0662-17-HMO. All data were fully anonymized; informed consent was waived according to the ethical committee’s instructions. The medical records of 83 (56 female, 27 male) OFP patients meeting our inclusion criteria attending the Orofacial Pain Clinic at the Hebrew University-Hadassah School of Dental Medicine between 2017 and 2018 were reviewed. Forty-three (28 female, 15 male) pain-free participants formed the control group.

Inclusion criteria: over 18 years of age, definite diagnosis of chronic OFP for at least 3 months according to the IHS or ICOP [17,48], and able to provide a saliva sample at a rate of at least 200 μL per 10 min. 

Exclusion criteria: background illnesses, including cancer or diseases affecting the salivary glands, such as Sjögren syndrome; alcohol use; patients who did not sign the consent form; and patients whose saliva sample was unusable, e.g., too foamy or bloody.

### 4.2. Orofacial Pain Diagnosis

The pain patients were divided based on their diagnoses:

Temporomandibular disorders (TMD) according to the diagnostic criteria for temporomandibular disorders (DC/TMD) [49], which are often associated with pain in the pre-auricular region and/or masticatory muscles, TMJ, and mandibular movement dysfunction. Primary headaches (also present in the facial area), including migraine (as well as facial migraine) and tension-type headache (TTH), according to the ICHD-3 [50]. Trigeminal neuralgia (TN) [51], which is characterized by recurrent unilateral brief electric-shock-like pains that are abrupt in onset and termination along the distribution of the trigeminal nerve and triggered by innocuous stimuli [48].Post-traumatic trigeminal neuropathy (PTN) [23], which involves unilateral facial or oral pain following trauma to the trigeminal nerve [48].Burning mouth syndrome (BMS), which is a chronic pain condition characterized by a moderate-to-severe sensation of burning from the oral mucosa, especially from the dorsum of the tongue with no clinical signs [48]. 

TN, PTN, and BMS formed the “neuropathic” group. Other diagnoses that were rare and non-specific (8 patients) formed the “others” group.

### 4.3. Collection of Data from Medical Records

Primary and resultant data were recorded on the standard intake form used in our clinic [52,53]. Demographic data included gender, age, and body mass index (BMI). Pain characteristics included onset (months) and distribution, which was charted by marking five areas on each side of the face. Patients were asked to report whether the pain was constant, came as an acute attack, or both, and were asked to rate the pain quality and intensity. Pain quality was assessed using the following descriptive terms: burning, electrical, pressure, throbbing, and stabbing sharp [54,55]. The intensity was rated by employing an 11-point verbal pain scale (VPS), where 0 was no pain and 10 was the worst imaginable pain. Health-related quality of life (HRQoL) over the last month on a 0–10 numeric scale was also recorded [56]. Patients were asked if their pain woke them from sleep. Clinical examination included masticatory apparatus palpation as previously described [57,58]. Intra-oral examination, including imaging, was performed to exclude dental, periodontal, and mucosal pathology. Brain and brainstem imaging were performed for TN to exclude intracranial pathology.

### 4.4. Saliva Collection

Unstimulated saliva was collected for 10 min as described previously [16,59] into pre-calibrated tubes. All 126 participants refrained from eating, drinking, and brushing their teeth 1 h prior to saliva collection. Patients did not take their medications, including sialagogues, before saliva collection. Volunteers rested for 10 min before saliva collection, sitting in an upright position in a quiet room, and were asked not to speak or leave the room until after the saliva was collected.

Saliva samples were centrifuged at 3500× g for 10 min at 2 °C to remove insoluble materials, cell debris, and food remnants. The supernatant fraction was aliquoted into polypropylene tubes and immediately stored at −80 °C. The running order of the samples was cycles of 2 samples from pain patients with the same diagnosis and then a control sample with the same gender and general BMI. 

### 4.5. eCB Purification

The extraction, purification, and quantification of saliva eCBs were performed using stable isotope dilution liquid chromatography/tandem mass spectrometry (LC-MS/MS) as previously described [60]. Briefly, total proteins were precipitated using ice-cold acetone and Tris buffer (50 mM, pH 8.0). Samples were then homogenized using a mixture of 0.5 mL ice-cold methanol/Tris buffer (50 mM, pH 8.0), 1:1, and 7 µL internal standard (22.4 ng d4-AEA). The homogenates were then extracted using ice-cold CHCl_3_:MeOH (2:1, vol/vol) and then washed with ice-cold chloroform three times. The samples were then dried under a thin stream of nitrogen and reconstituted in MeOH. Analysis using LC-MS was performed on an AB Sciex (Framingham, MA, USA) Triple Quad 5500 Mass Spectrometer and a Shimadzu (Kyoto, Japan) UHPLC System, while the liquid chromatographic separation was acquired via a Kinetex (Phenomenex) column (C18, 2.6 mm particle size, 100 × 2.1 mm). Sample levels of AEA, 2-AG, AA, PEA, and OEA were measured against standard curves and then expressed as fmol/mg protein.

### 4.6. Statistics

SPSS version 25 software was used for all calculations. To examine the differences in eCB levels for nominal and categorical background variables, t-tests and one-way analysis of variance were performed, and when significant differences were found, additional post hoc Scheffe tests were performed. 

A Spearman coordinator was used to examine the specific categories that made up the differences. The differences between the eCB types and specific background variables were examined for each diagnosis using the Kruskal–Wallis test.

## 5. Conclusions

Our findings suggested that the ECS played a significant role in the pathogenesis of OFP disorders. In addition, the non-invasively gathered salivary samples exhibited signature eCB profiles for prominent OFP disorders. Therefore, the profile of salivary eCBs and eCB-like molecules may be used as biomarkers to aid in the diagnosis and management of these patients. Future research is needed to evaluate whether salivary eCB levels correlate with serum levels in chronic OFP disorders, to understand the underlying mechanisms of altered ECS found in these patients, and to evaluate the therapeutic use of cannabinoids possibly as part of a personalized medicine approach for the management of chronic OFP disorders.

## Figures and Tables

**Figure 1 ijms-23-13017-f001:**
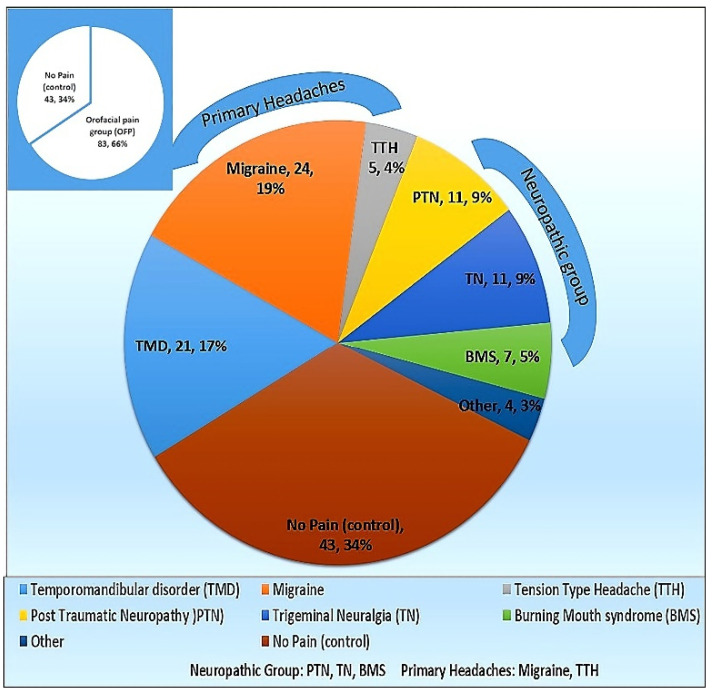
Cohort distribution according to the diagnosis.

**Table 1 ijms-23-13017-t001:** Salivary eCB levels in primary headache and neuropathic pain groups compared with the controls (mean ± SD; fmol/mg protein).

Group *	eCBs	Pain	Control	*p*-Value
**Primary headache group** **(migraine and TTH)**	**AEA**	0.09 ± 0.09	0.17 ± 0.18	**0.02**
	**OEA**	35.65 ± 40.31	67.7 ± 77.09	**0.04**
	**AA**	1354.3 ± 1386.9	2327.15 ± 2539.3	0.06
**Neuropathic group** **(PTN, TN, and BMS)**	**2-AG**	35.97 ± 42.56	54.71 ± 36.47	**0.05**

AEA, OEA, and AA levels were lower in the migraine sufferers than the controls (*p* = 0.02, *p* = 0.04, and *p* = 0.06, respectively). 2-AG was lower in the neuropathic group than in the controls (*p* = 0.05). * Grouped according to etiology and characteristics as previously reported [17,18]. Non-parametric Mann–Whitney U test.

**Table 2 ijms-23-13017-t002:** Salivary eCB levels for migraine, TTH, TN, and BMS compared with the controls (mean ± SD; fmol/mg protein).

Diagnosis	eCBs	Pain	Control	*p*-Value
**Migraine**	PEA	10.77 ± 11.76	12.92 ± 11.75	**0.05**
**TTH**	2-AG	16.85 ± 6.56	54.71 ± 36.47	**0.02**
**TN**	2AG	12.71 ± 14.07	54.71 ± 36.47	**<0.001**
**BMS**	AEA	0.56 ± 0.87	0.17 ± 0.18	**0.01**

Lower levels of PEA were found for migraine and 2-AG for TTH and TN compared with the controls (*p* = 0.05, *p* = 0.02, and *p* < 0.001 respectively) and higher levels of AEA were found for BMS compared with the controls (*p* = 0.01). Non-parametric Mann–Whitney U test.

**Table 3 ijms-23-13017-t003:** Patient and pain characteristics in relation to eCB levels for PTN, TN, and migraine.

Post-Traumatic Neuropathy (PTN)					
	eCBs		N	Mean ± SD (fmol/mg)	*p*-Value
**Gender**	**AA**	M	5	2547.9 ± 1662	**0.017**
		W	6	613.2 ± 449	
**HRQoL (0** **→** **10)**	**AA**		6	** 0.926	**0.008**
**Trigeminal Neuralgia (TN)**					
**Waken**	**AEA**	No	7	0.08 ± 0.13	**0.033**
		Yes	3	0.43 ± 0.23	
	**OEA**	No	7	16.55 ± 12.54	**0.017**
		Yes	3	200.06 ± 125.08	
	**PEA**	No	7	4.8 ± 4.08	**0.017**
		Yes	3	38.42 ± 21.51	
**Pain onset (months)**	**AA**		10	0.726 *	**0.017**
**Migraine**					
**Gender**	**OEA**	M	4	93.4 ± 73	**0.006**
		W	15	25.9 ± 15	
	**PEA**	M	4	25.31 ± 20	**0.004**
		W	15	6.9 ± 4	
**BMI**	**2-AG**	>30	6	52.8 ± 53	**0.046**
		<30	13	34.3 ± 50	
**VPS (0** **→** **10)**	**OEA**		19	−0.582 **	**0.009**
	**PEA**		19	−0.470 *	**0.042**

For PTN, AA was significantly lower in women and was also positively correlated with HRQoL. For TN, AA was positively and significantly correlated with pain onset. AEA, OEA, and PEA correlated with sleep awakenings (due to pain). In the migraine group, OEA and PEA were significantly lower in women. 2AG was significantly higher in patients with BMI >30, while OEA and PEA were negatively correlated with VPS. * Correlation was significant at the 0.05 level (two-tailed). ** Correlation was significant at the 0.01 level (two-tailed). Kruskal–Wallis test.

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
