# Peer review of "Salivary Endocannabinoid Profiles in Chronic Orofacial Pain and Headache Disorders: An Observational Study Using a Novel Tool for Diagnosis and Management"

_ijms, 2022, doi:10.3390/ijms232113017_

Round 1

Reviewer 1 Report

The scientific paper "Salivary endocannabinoid profile in different chronic orofacial pain disorders: A tool for diagnosis and management” aimed to investigate the quality and quantity of specific endocannabinoids (eCB) compounds in distinct chronic orofacial pain (OFP) disorders and to evaluate their efficacy as measurable biomarkers of these pain disorders. Furthermore, aimed to evaluate the influence of different clinical parameters and pain characteristics on salivary eCB levels in distinct chronic OFP disorders. I can make the following considerations:

1)       Increment the conclusions in the abstract;

2)       Review the entire manuscript as there are several typos and missing spaces. Ex: 1. Introdction; anandamide(AEA) etc;

3)       Due to the great importance of the anatomical knowledge of the temporomandibular joint (TMJ) and trigeminal nerve for the research carried out and for the readers, I recommend inserting a paragraph in the introduction with the main elements of the TMJ, location of the trigeminal ganglion and its divisions;

4)       Bibliographic references are not in the correct format indicated for the IJMS journal. Please correct;

5)       The results were very poorly written. I recommend completely redoing. The authors limited themselves to inserting figures and tables, without describing the results in the form of text. The caption of figure 1 does not portray the entirety of the research participants. It needs to be redone and bring more elements, preventing the reader from reading back to the text for understanding;

6)       The lack of numbering in the lines of the submitted file makes it difficult to describe the corrections that must be made in the manuscript. For example, in the discussion there is a very long paragraph, which starts on page 7 and takes up almost all of page 8. Please adjust;

7)       Insert the limitations of the study at the end of the discussion.

Author Response

A letter to the reviewer is attached 
Please refer also to the attached track changes version of the manuscript.

Reviewer 2 Report

Dear Authors,

I reviewed the article entitled “Salivary endocannabinoid profile in different chronic orofacial pain disorders: A tool for diagnosis and management”.

The article is scientifically sound, well written, and provides useful tools for research into a future with personalized cannabinoid treatment for different types of pain.

Major revisions:

1. The title can be improved. This is for the main part an observational study, therefore this aspect should be included as it is of importance.

2. The patient’s inclusion and exclusion criteria should be expressed in greater detail.  Especially the general treatment recommended and used, as well as allopathic or holistic remedies possibly used by patients, general diet ques, etc.

3. The Results section needs to be reorganized as is presented quite confusingly. Information about tables should be presented above each corresponding one.

4. The study has its limitations and can possibly be considered a preliminary evaluation, but the fact that the authors signal these limitations is worth appreciating.

Good luck!

Author Response

A letter to the reviewer is attached

please also refer to the an attached track changes version of the manuscript

Round 2

Reviewer 1 Report

1) The callout in the manuscript text of the references are not in the IJMS MDPI standard. Ex: [1]

2) Text must be in justified alignment format

Author Response

Reviewer 1

1) The callout in the manuscript text of the references are not in the IJMS MDPI standard. Ex: [1]
Answer 
Thank you, corrected according to MDPI - IJMS instructions 

2) Text must be in justified alignment format
Answer 
Corrected as requested

Thank You!

Reviewer 2 Report

Dear authors, 

The manuscript has been significantly improved and now the importance of your work is highlighted and explained properly. Congratulations! 

Author Response

reviewers 2
The manuscript has been significantly improved and now the importance of your work is highlighted and explained properly. Congratulations! 

Answerr
Thank you!